# Understanding and including 'pink-collar' workers in employment-based travel demand models

Yiping Yan[1,2,3]*, Abraham Leung[1], Matthew Burke[1], James McBroom[4]

**1** Cities Research Institute, Griffith University, Brisbane, Queensland, Australia, **2** School of Engineering and Built Environment, Griffith University, Brisbane, Queensland, Australia, **3** School of Vehicle and Mobility, Tsinghua University, Beijing, China, **4** School of School of Environment and Science, Griffith University, Brisbane, Queensland, Australia

* yan_yiping@mail.tsinghua.edu.cn

**Data Availability Statement:** All data files are available from the Queensland Government Open Data database: https://www.data.qld.gov.au/dataset/queensland-household-travel-survey-

## Abstract

The segmentation of commuters into either blue or white-collar workers remains is still common in urban transport models. Internationally, models have started to use more elaborate segmentations, more reflective of changes in labour markets, such as increased female participation. Finding appropriate labour market segmentations for commute trip modelling remains a challenge. This paper harnesses a data-driven approach using unsupervised clustering–applied to *2017–20 South East Queensland Travel Survey* (SEQTS) data. Commuter types are grouped by occupational, industry, and socio-demographic variables (i.e., gender, age, household size, household vehicle ownership and worker skill score). The results show that at a large number of clusters (i.e., k = 8) a highly distinct set of commuter types can be observed. But model run times tend to require a much smaller number of market segments. When only three clusters are formed (k = 3) a market segmentation emerges with one female-dominated type ('pink collar'), one male-dominated type ('blue collar') and one with both genders almost equally involved ('white collar'). There are nuances as to which workers are included in each segment, and differences in travel behaviours across the three types. 'Pink collar' workers are mostly comprised of female clerical and administrative workers, community and personal service workers and sales workers. They have the shortest median commutes for both private motorised and active transport modes. The approach and methods should assist transport planners to derive more accurate and robust market segmentations for use in large urban transport models, and, better predict the value of alternative transport projects and policies for all types of commuters.

## Introduction

The first large strategic travel demand models of the 1950s and 60s, such as the *Chicago Area Transportation Study* in 1959 analysed home-based commute trips by splitting workers into two types: 'blue-collar' industrial workers and 'white-collar' office workers. Whilst somewhat

series/resource/351cd3dd-939b-43bb-9e88-37cb0cb20c82.

**Funding:** YY is the recipient of a Griffith University International Postgraduate Research Scholarship. https://www.griffith.edu.au/research-study/scholarships/guiprs AL is funded by the Advance Queensland Industry Research Fellowship (2021 round) and the Transport Academic Partnership (Queensland Department of Transport and Main Roads, The Motor Accident and Insurance Commission) and Transport Innovation and Research Hub (Brisbane City Council). https://advance.qld.gov.au/industry-research-fellowships-recipients https://www.tmr.qld.gov.au/TransportAcademicPartnership MB receives funding from the Australian Research Council, the Queensland Department of Transport and Main Roads, the Motor Accident and Insurance Commission, Brisbane City Council, and the City of Gold Coast. Matthew is a member of the Queensland Government's Fares Advisory Panel, Cycling Advisory Group and Bus Safety Forum, the Brisbane Lord Mayor's Transport Strategy External Advisory Group, and the City of Gold Coast's Active Transport Committee. Matthew is a member of scientific committees with the Australasian Transport Research Forum, the Eastern Asia Society for Transportation Studies and the Transportation Research Board of the US National Academy of Sciences. https://www.tmr.qld.gov.au/TransportAcademicPartnership JB has not received funding from external sources. The funders had no role in study design, data collection and analysis, decision to publish, or preparation of the manuscript.

**Competing interests:** The authors have declared that no competing interests exist.

acceptable in the 1950s, over time this blue collar/white collar duality become less and less representative of the labour force. Economic changes produced greater differentiation in the types of employment offered in increasingly knowledge-based cities. Women began to participate much more in the workforce, and commute, in greater numbers. Travel behaviour studies began to focus on women's travel from many perspectives, including: gender differences in distances to work; mode of travel; automobile occupancy; and, the propensity to combine multiple destinations in one trip [1–4]. Gender was soon recognised as one of the key socio-demographic influences on commuting behaviour. Women and men who worked in the same occupational category often had different commuting patterns [5]. Nevertheless, many transport models for cities/regions, including Australian transport models, tended to subsume women's travel by retaining a simplistic blue/white collar market segmentation [6–11]. Cities such as London [12,13] and Paris [14–16] in Europe, and many regional models in North America [17–19] have similar blue/white collar market segmentations in their transport models. But more nuanced and differentiated market segmentations are being developed. This paper reports on a novel typology of workers developed for South-East Queensland, Australia, that can help direct this field towards improved market segmentations that better represent the structure of today's workforce. Instead of the 'traditional approach' of experts deductively searching through travel survey data to identify preferred segmentation of commuters, a data-driven clustering approach was used to explore what different segments emerge when one allows only three clusters, or as many as eight clusters, from the same dataset. As will be shown, this unsupervised clustering analysis produced a more nuanced understanding of workers' occupational and socio-demographical characteristics. These understanding should help modellers better represent commuting by all workers, including women.

## Background

### Evolution of the labour force

During much of the twentieth century, employment was viewed as full-time and permanent waged, where a male was the main income earner of a (hetero-normative) household and a female was the main domestic carer [20]. Since the latter part of the Twentieth Century, journey-to-work patterns have transformed due to changes in the nature of employment, transportation costs, economic shifts, increasing female workforce participation and public policies that increasingly favoured labour mobility and uncertainty [21,22]. The rise of the 'gig economy' and telecommuting workers [23] the expansion of the service industry, and a decline in manufacturing in the West, all added more complexity. Strategic transport models were often only partially updated, usually in piece-meal fashion, to represent the commuter behaviours that emerged.

The term 'blue collar' to identify manual labouring workers first appeared at the beginning of the 20th century; it differentiates from the 'white collar' that identifies a class of administrative workers [24]. The early mid-century modelling pioneers harnessed these two employment types, and tried to forecast commuting flows for these workers, to help predict peak-hour travel flows, and, in turn, how much road and public transport infrastructure was needed. They did so noting the mode choices of these workers, their travel times, and their trip movements, withing commuter's observed spatial and temporal constraints. This developed these understandings from the first household travel surveys used to develop and calibrate these models. In this immediate post-World War II period, economies in the US, UK and Australia were dominated by manufacturing, mining and industry [25]. With high tariffs on imported goods, firms hired workers in steelworks, automobile and chemical plants, and a range of other factories. With low female workforce participation, and with limited computing power, a simple white/blue collar split was a reasonable approach for transport modellers to employ.

A series of structural changes in the 1970s and 1980s saw a shift towards a post-industrial future. Reaganomics in the USA, Thatcherism in the UK and the Hawke/Keating reforms in Australia all saw tariffs reduced, state utilities such as airlines and airports sold off, and the economy increasingly liberalised. Western nations began to import more manufactured goods than they made locally. Machines replaced many workers in the factories and then in the mines. Knowledge work, including in finance, banking and education, and services work, including in health, began to increase as the total number of employees in manufacturing and labouring fell. At the same time, women's liberation movements achieved increasing equality in access to work, and in pay rates. Discriminatory practices were outlawed and/or discouraged. Childcare services expanded. Many households obtained two or more cars. It became a different world.

The workers that today form the 'blue-collar' and 'white collar' market segments of the labour force differ from those of previous eras. Workers in many Australian or US factories may have more advanced skills and higher education training than the workers of the past. Today's workers may have many different employers over a lifetime and be increasingly flexible in terms of their work arrangements. The previously unionised tradesmen (electricians, plumbers, etc.) who worked for big construction firms or utilities in the 1960s are today mostly self-employed contractors, effectively small business owners whose key assets are their labour and their skills. Within the 'white-collar' segment we now see 'portfolio workers', who trade on their knowledge and may work as contractors to multiple employers, perhaps based from a co-work space (shared by multiple micro-firms rather than one big employer) in the inner city. The 'gig-economy' of smart-phone apps (Uber, Deliveroo, etc.) has created a new class of 'platform' worker, with minimal labour protections [26]. To classify all these worker types using the 1960s division of 'blue collar' and 'white collar' doesn't seem appropriate. But again, there has been little analysis of these trends, or of what a more robust market segmentation might look like for commuter travel demand modelling.

To focus on labour market segmentation alone is not to ignore other changes for these workers. Spatially, middle- and upper-income workers have reclaimed once industrial inner-city areas, including waterfronts, pushing lower-income workers to the city fringe. Inner-city workers in a particular profession can have very different attitudes and perceptions to their outer-suburban peers. Inner-city gentrifiers tend to express more environmentally and socially progressive attitudes and a preference for urban lifestyles and amenities, which suggests that they may engage in more sustainability-conscious behaviours, including in travel [27–30]. The rise of inner-city cycling (with Washington D.C. going from 1% to 5% cycling rates in the last decade) is partly seen as a result of in-migration by knowledge workers. Numerous studies have indicated that residential context, socio-economic characteristics like employment type, and environmental attitudes are likely to affect commuters' behaviour [31]. It is not the main intention of this paper to study such socio-demographic changes in-depth; these are future research agendas. But one needs ways to keep transport models representative of cities and their citizens.

## Employment market segmentation in transport models

The 1955 *Detroit Metropolitan Area Traffic Study* [32] crudely took ratios of workers in industrial plants per 1,000 residents, and ratios of workers located within the core area of the central business district per 1,000 residents to represent 'blue' and 'white' collar workers (albeit without actually using those terms). The field advanced and by 1965 the *Brisbane Transportation Study* adopted a more detailed classification of the employment market, by grouping such industries as primary production, manufacturing, building and public services, business services and commerce, public authorities and professional services, personal services and other

industries [33]. Despite all the changes in labour force markets during the last 60 or more years, the *Brisbane Strategic Transport Model* (BSTM), the successor of that work, is still using 'blue/white collar' market segmentation [10]. Larger cities with more complex commuter markets have moved on, with Sydney's model today including 'pink-collar' workers to represent female-dominated service jobs, and 'gold-collar' workers to represent advanced business and finance industry jobs [34].

The use of deductive approaches to identify and analyse a 'pink-collar' market segment in household travel surveys, and include this in transport modelling, is becoming more common, for good reason. There are many constraints on women's travel, often due to childcare and other family care responsibilities, and due to different motor vehicle availability. This limits their work trip distances and commuting mode preferences [35,36]. Nevertheless, studies about how to incorporate these gender differences in mode choice models and other parts of the large strategic transport models, are rarer. Given the heavy dominance of men in the transport planning/engineering field, the potential for unconscious male-bias when using deductive approaches is probably high. That is, male modellers are not deliberately choosing sub-optimal classifications when searching for and analysing commuters in female-dominated professions. It's that they may unconsciously bring biases to their choices and accidentally err. It's relatively easy to assume female workers in a particular occupation will have similar commuting behaviours as men, when they do not; or vice versa. For this reason, inductive data approaches are generally preferable for such tasks, as they remove or reduce the potential for researcher biases.

## Approach and methods

### Data collection

This paper uses data from the *South East Queensland Travel Survey 2017–2020* (SEQTS), made publicly available in an open data portal by the Queensland Department of Transport and Main Roads (DTMR) [37]. The SEQTS data used a stratified, multi-stage clustered sampling technique. 101,616 trips were recorded from 36,264 respondents living in 14,715 households, with a response rate of about 50%. Travel diaries were completed for all members of the household aged five and over, capturing information on all trip stages for each trip. More information on the data collection process is available from DTMR [37].

A 'main mode' was allocated to each trip by the DTMR, including journey-to-work trips. The 'main mode' is the mode used on any trip stage in an overall trip, using a hierarchy (from high to low) of public transport modes, then private vehicle travel, then cycling, then walking. If a trip is made by both car and public transport it is categorised as public transport, and so on. The sample included 9,150 workers' (56% males and 44% females) with journey-to-work trip records, with five cases excluded due to abnormally large commutes (>200km that indicate travelling outside the region). Percentages of employed persons by occupation in our sample, and in the region's comparable census data, are provided in Table 2. The comparison between gender split by occupation in our sample and in the census data are shown in Fig 1. As presented below, the dataset adequately reflects the actual commuters of the region, albeit with some under- and over-representation of women in a few categories in the SEQTS data sample. The dataset we adopted was collected before the Covid-19 pandemic, and is not 'polluted' by the forced telecommuting seen in the region for much of 2020–21.

### Selection of parameters in the market segmentation model

Standard occupation classifications provided in census data have been commonly used by transport modellers. The underlying assumption of market segmentation is that people with

**Table 1. Percentage of different occupations in our data sample vs. ABS census data for the region.**

| Occupation type | Sample size | Percentage by occupation | |
|---|---|---|---|
| | | (Sample data, 2017–19) | (ABS Census data, 2016) |
| Managers | 1,247 | 13.5% | 12.0% |
| Professionals | 2,313 | 25.0% | 21.7% |
| Community and Personal Service Workers | 955 | 10.3% | 11.2% |
| Clerical and Administrative Workers | 1,212 | 13.1% | 14.3% |
| Sales Workers | 650 | 7.0% | 10.0% |
| Technicians and Trades Workers | 1,505 | 16.2% | 13.6% |
| Machinery Operators And Drivers | 555 | 6.0% | 5.9% |
| Labourers | 712 | 7.7% | 9.7% |
| Miscellaneous | 118 | 1.3% | 1.6% |

different characteristics place different importance on different aspects of service [38,39]. Employees' occupational and socio-demographic characteristics should play a significant role in impacting their commuting behaviour. In Australia the *Australian and New Zealand Standard Classification of Occupations* (ANZSCO 2013, Version 1.2) has 1,023 listed occupations classified into 8 major groups: managers, professionals, technicians and trade workers, community and personal service workers, clerical and administrative workers, sales workers, machinery operators and drivers, and labourers [40], as shown in Table 1.

As shown in Table 2, ANZSCO Skill Level is a skill-based classification used to classify all occupations and jobs, and group them into successively broader categories for statistical and

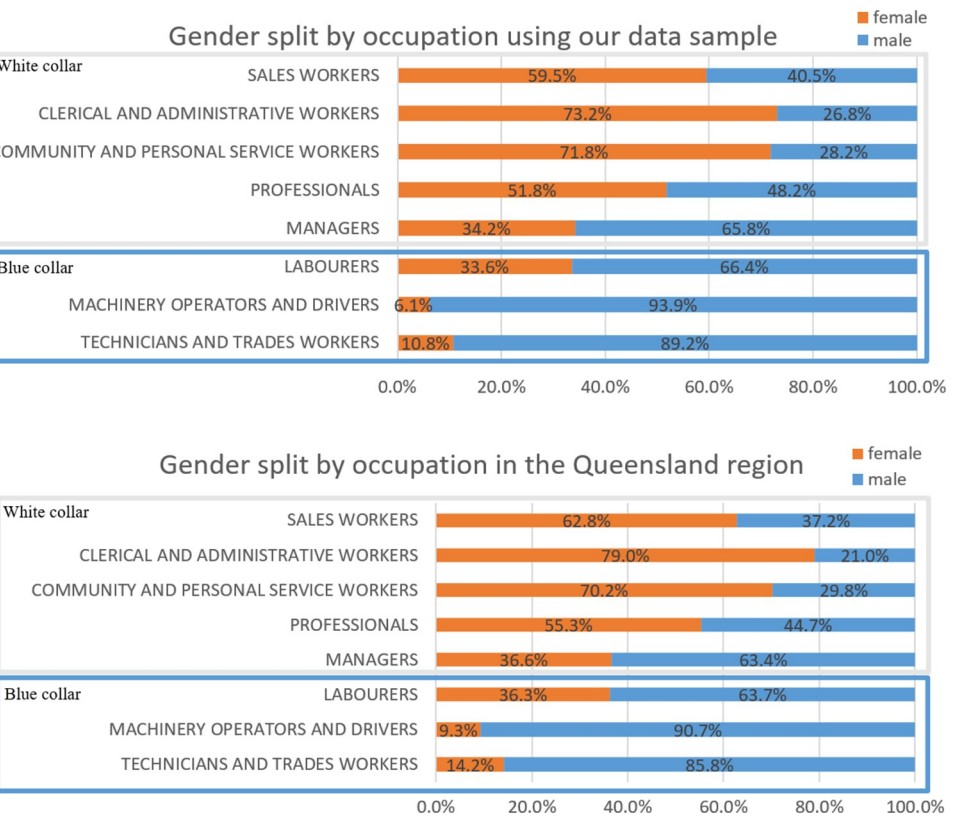

**Fig 1. Gender split by occupations (our data sample vs. ABS data in the region).**

**Table 2. Definition of blue/white collar by ANZSCO Level 1 major groups.**

| ANZSCO Level 1 (Major Groups) | 'Collars' in BSTM | ANZSCO Level 2 (Sub-Major Groups) | ANZSCO Level 3 (Minor Groups) | ANZSCO Level 4 (Unit Groups) | ANZSCO Level 5 (Occupations) |
|---|---|---|---|---|---|
| 1. Managers | White | 4 | 11 | 38 | 99 |
| 2. Professionals | White | 7 | 23 | 100 | 318 |
| 3. Technicians and Trades Workers | Blue | 7 | 21 | 66 | 179 |
| 4. Community and Personal Service | White | 5 | 9 | 36 | 105 |
| 5. Clerical and Administrative Workers | White | 7 | 12 | 33 | 80 |
| 6. Sales Workers | White | 3 | 5 | 19 | 37 |
| 7. Machinery Operators and Drivers | Blue | 4 | 7 | 22 | 77 |
| 8. Labourers | Blue | 6 | 9 | 44 | 128 |
| Total Number of classifications: (8) | (2) | (43) | (97) | (358) | (1023) |

other types of analysis based on the similarity of their attributes in the Australian and New Zealand labour markets. Using aspects of both skill level and skill specialisation, 'sub-major' groups are then collected into eight 'major' groups. ANZSCO1 (Level One) represents the broadest level of ANZSCO with 8 major groups and ANZSCO3 (Level Three) represents for minor groups subdivided from higher level groups. ANZSCO classifies occupations according to two criteria–skill level and skill specialisation. In this study, the skill specialisation for ANZSCO3 is adopted as it is more comprehensive. Skill Levels, ranging from 1 (highest) to 5 (lowest), is measured by the level of formal education and training, previous experience, and/ or on-the-job training, required to competently perform in that occupation. In the current BSTM, the major groups are crudely aggregated into two main types to form the 'white/blue collar' market segmentations used in the mode choice model (see Table 1).

Unsupervised clustering analysis is an alternative approach to use such data to generate an alternative market segmentation. In essence, it reveals subgroups within heterogeneous data, where each individual cluster has greater homogeneity than the whole [41]. The methods selected for this work consisted of three major stages: 1) data preparation, including feature selection, and extraction; 2) use of an unsupervised PAM (partition around medoids) clustering algorithm to analyse datasets made of mixed-typed data; and, 3) comparing results for different numbers of clusters (k = 2, 3, 4, 5, 6, 7, 8) to help identify an optimal k-value using an average silhouette width measure [42].

The selection of variables for the cluster analysis was based on data availability within the SEQTS and guided by prior research in the field [43–45]. Variables that were too similar were selectively eliminated to avoid co-linearity. The socio-demographic characteristics finally included were: gender, age, occupation groups (ANZSCO Classification of Occupations), Skill Level, industry type, household size, household vehicle ownership, and household bicycle ownership. This approach did omit some variables known to influence travel behaviour. The SEQTS itself does not have any attitudinal data, such as environmental attitudes or attachment to particular modes like driving. This is a major limitation common to most other HTS datasets around the world.

## Unsupervised Partitioning Around Medoids (PAM) clustering

Unsupervised methods have an advantage over traditional clustering methods (such as hierarchical) as they do not require prior knowledge and are less subjective. Also, compared to the more commonly used k-means clustering algorithm, Partitioning Around Medoids (PAM) is

more intuitive and robust to noise and outliers in the underlying data, due to the properties of distances being used. PAM is also capable of analysing mixed-type data, where numeric, nominal, or ordinal features coexist. The main disadvantage of PAM is its unsuitability for clustering non-spherical (arbitrary shaped) groups of objects, but this became irrelevant as the clustering formed relatively spherical shapes.

Increasing the number of clusters raises the risk of overfitting, by definition. We limited our analysis to a maximum of eight clusters (k = 2 to 8) and calculated the silhouette coefficients for each k to qualify the relevancy of the chosen number of clusters from a statistical perspective. In distance-based clustering of mixed-type data, a good performance of detecting clusters can be achieved using Gower's dissimilarity followed by PAM [46,47]. The Gower distance metric was used to measure proximity or similarity across individuals within our dataset. Most simply, Gower distance is computed as the average of partial dissimilarities across individuals, in which each numeric-valued feature is standardised, and the distance is calculated as the average of all feature-specific distances. However, if variables are of mixed (qualitative as well as numeric) types, partial dissimilarity is calculated differently. For numeric features, it is computed as the ratio between absolute difference of observations $X_i$ and $X_j$ and maximum range observed from all individuals (thus scaling all dissimilarities to lie between 0 for identical, and 1 for maximally dissimilar):

$$d_{ij}^{(f)} = \frac{|x_i - x_j|}{|x_{MAX} - x_{Min}|}$$

The Gower distance's formula is provided below with n representing the sample size:

$$d(i,j) = \frac{1}{n} \sum_{k=1}^{n} d_{ij}^{(f)}$$

As a classical partitioning technique of clustering, a K-medoid algorithm is applied to cluster the dataset into k clusters fixed a priori, incorporating the Gower distance metrics result. Silhouette coefficient is used, and a high value of that index implies well clustered groups [48]. In this study, Silhouette width particularly measures whether workers that have similar socio-economic traits and occupation type to each other are placed in the same cluster and whether clusters are tightly bound with a substantial distance from each other.

In Fig 2, when k = 4, the data points are not compact within the cluster to which they belong and there is more overlapping between clusters; this is sub-optimal. It seems that increasing k value to 8 only insignificantly improve the width result from k being 2 or 3. Considering the result from average Silhouette width measure, as well as the significant overall model run-time savings of having fewer market segments in strategic transport models, the most useful number of clusters for the SEQTS dataset was just three.

All the clustering analysis was performed using *R* software. Once market segments were identified, the travel behaviour of these groups was then explored using *Python* with the same SEQTS dataset. The variables calculated included: mode shares; and, trip distances by mode, with the latter represented as violin plots for ease of interpretation.

## Results

### Occupational clusters when k = 8 and when k = 3

Tables 3 and 4 show the occupational clusters revealed when there were eight clusters (k = 8) and when there were only three (k = 3), respectively. These serve difference purposes, as the k = 8 solution is more useful for refined models. As the number of clusters reduces, one can

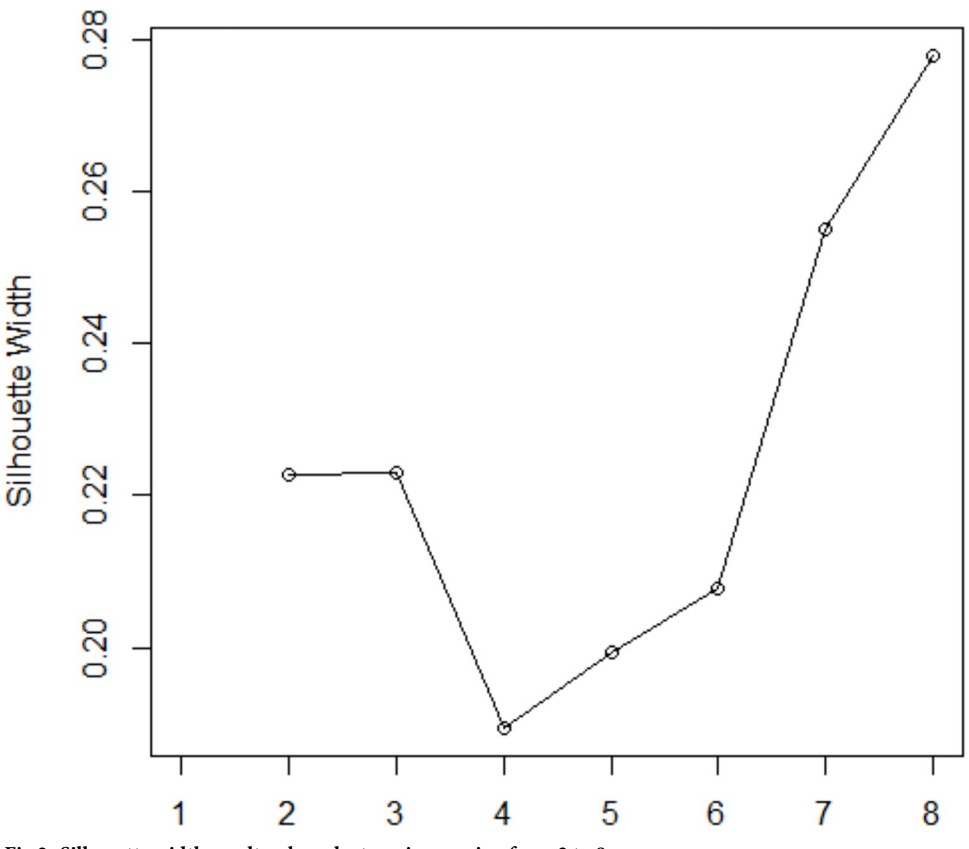

**Fig 2. Silhouette width results when clusters size ranging from 2 to 8.**

see the merging of these groups in interesting ways, and how the unstructured classification process shifts commuters into different groups when forced to place them in a limited number of bins (i.e., the transition from k = 8 to k = 3).

Interrogating these results one can see that at k = 8, the disaggregated commuter types include the more traditional 'blue collar' (Groups 3, 5 and 8), and 'white collar' (Groups 4, 6, 7) groupings. But there are also two other distinct female-dominated clusters: community and professional service workers (Group 1), and clerical/administrative workers mostly across the retail and health industries (Group 2). Neither of these clusters fits either the 'white' or 'blue' collar categories. Interestingly, when aggregated to just six clusters (k = 6, not shown in this paper), these mostly female community and personal service workers, and clerical and administrative workers stay almost the same. When further aggregated to only four clusters (k = 4, not shown in this paper), the two disaggregated female-dominated clusters merge into one single cluster.

When forced into just three clusters (k = 3), the male-dominated professional workers group and the female-dominated professional workers group seen in Table 2 seem to merge into one cluster, the 'white-collar' group 3 in Table 4. 64% of its members are female. There is also female-dominated cluster (86.2% female) which we refer to as 'pink-collar'. Third, there is a 'blue-collar' cluster, 98.2% of whom are male. Details about the median/average value of socio-demographic characteristics and mode share/median trip distance by mode of all these groups (when k = 8 and 3) are summarised in Tables 3 and 4, respectively.

The t-Distributed Stochastic Neighbour Embedding (t-SNE) method is a specialised technique for reducing dimensionality, making it especially effective for visualising high-

**Table 3. Summary results at highest level of disaggregation (k = 8).**

| Cluster No. | 1 | 2 | 3 | 4 | 5 | 6 | 7 | 8 |
|---|---|---|---|---|---|---|---|---|
| Dominant Age Group | 10 (45–49 years) | 10 (45–49 years) | 10 (45–49 years) | 10 (45–49 years) | 9 (40–44 years) | 9 (40–44 years) | 10 (45–49 years) | 8 (35–39 years) |
| Dominant Sex (Male/Female) | Female (931/1017) | Female (1282/1360) | Male (735/755) | Male (841/998) | Male (1453/1536) | Male (1166/1166) | Female (1373/1373) | Male (769/944) |
| Dominant ANZSCO1 (count) | Community and professional service workers (755) | Clerical and administrative workers (869) | Machinery operators and drivers (532) | Managers (964) | Technicians and trades workers (1425) | Professionals (1114) | Professionals (1199) | Labourers (626) |
| Dominant Industry (count) | Health (490) | Retail (453) | Transport (405) | Construction (216) | Construction (582) | Education (226) | Education (475) | Other (347) |
| Household size (median/mean) | 3/3.016 | 2/2.740 | 3/3.057 | 3/3.068 | 3/3.083 | 3/3.068 | 2/2.805 | 3/3.059 |
| Household vehicle number (median/mean) | 2/2.293 | 2/2.242 | 2/2.291 | 2/2.354 | 2/2.345 | 2/2.148 | 2/2.183 | 2/2.144 |
| Bike ownership (median/mean) | 1/1.439 | 1/1.239 | 1/1.334 | 2/1.934 | 1/1.553 | 2/1.916 | 1/1.535 | 1/1.314 |
| Workplace is in Central Business District (CBD) (mean) | 3.8% | 11.8% | 3.8% | 11.9% | 4.95% | 19.5% | 11.5% | 5.6% |
| Skillscore (median/mean) | 4.0/3.641 | 3.5/3.663 | 4.0/4.054 | 1.50/1.434 | 3.0/2.884 | 1.0/1.096 | 1.0/1.10 | 4.5/4.603 |
| Mode share of active transport* | 3.44% | 1.7% | 0.7% | 1.4% | 1.6% | 5.6% | 2.7% | 3.4% |
| Mode share of private motorised vehicle | 88.4% | 84.1% | 94.2% | 88.5% | 93.9% | 76.2% | 83.2% | 89.8% |
| Mode share of public transport | 7.77% | 13.7% | 4.6% | 9.9% | 4.5% | 18.0% | 13.7% | 6.5% |
| Median trip distance (km) of active transport | 0.93 | 1.23 | 2.11 | 5.90 | 1.94 | 4.54 | 1.54 | 1.285 |
| Median trip distance (km) of private motorised vehicle | 10.97 | 13.115 | 19.53 | 17.72 | 18.795 | 16.005 | 13.05 | 14.255 |
| Median trip distance(km) of public transport** | 20.61 | 22.79 | 18.67 | 24.45 | 26.62 | 19.335 | 18.71 | 20.490 |

*Active transport includes journeys made by walking only, or by cycling, but does not include public transport trips where these modes were used for the first or last mile.

**Median public transport trips being longer than the private car trips was unexpected. However, Brisbane's express busway and rail networks carry many suburban commuters into the centre; there are many inter-city commute trips made, including between the Gold Coast, Ipswich, the Sunshine Coast and Brisbane; there are almost no cross-suburban public transport routes, requiring transfers in Brisbane's CBD; and, the city's road networks tend to offer more direct routes between Origin-Destination pairs, and lesser trip distances, than the rail corridors.

dimensional data within a low-dimensional space (e.g., two- or three-dimensional) while preserving local structure [49]. Using t-SNE, the data can be visualised into a two-dimensional plot (Fig 3) showing the clusters of the blue, white and pink collar k = 3 solution as presented in Table 4. It should be noted that this is visualisation technique to tell how far apart the clusters are in a low-dimensional, they not meant for quantitative interpretation.

Table 4. Summary results at three levels of disaggregation (k = 3).

| Cluster | 1 (Blue Collar) | 2 (Pink Collar) | 3 (White Collar) |
|---|---|---|---|
| Dominant Age Group | 9 (40–44 years) | 9 (40–44 years) | 10 (45–49 years) |
| Dominant Sex (Male/Female) | Male (3732/3801) | Female (2060/2391) | Female (1892/2957) |
| Dominant ANZSCO1 (count) | Technicians and trades workers(1408/3801) | Clerical and administrative workers (1049/2391) | Professionals (2231/2957) |
| Dominant Industry (count) | Construction (1090/3801) | Retail (605/2391) | Health (928/2957) |
| Household size (median/mean) | 3/3.1 | 3/2.9 | 3/2.9 |
| Household vehicle number (median/mean) | 2/2.3 | 2/2.3 | 2/2.2 |
| Bike ownership (median/mean) | 1/1.7 | 0/1.2 | 1/1.6 |
| Skillscore (median/mean) | 3/3.0 | 4/3.9 | 1/1.3 |
| Workplace is in Central Business District (CBD) (mean) | 6% | 9% | 14% |
| Mode share of active transport | 63/3793 (1.7%) | 61/2379 (2.6%) | 111/2947 (3.8%) |
| Mode share of private motorised vehicle | 3512/3793 (92.6%) | 2045/2379 (86.0%) | 2400/2947 (81.4%) |
| Mode share of public transport | 218/3793 (5.7%) | 273/2379 (11.5%) | 436/2947 (14.8%) |
| Median trip distance(km) of active transport | 2.08 | 1.18 | 2.89 |
| Median trip distance(km) of private motorised vehicle | 18.405 | 12.28 | 13.75 |
| Median trip distance(km) of public transport | 25.765 | 21.07 | 19.225 |

## Profile of the three types of commuters

Fig 4 provides further occupational breakdown of the three clusters (k = 3). The 'pink-collar' group is mainly comprised of female clerical and administrative workers (35.85%), community and personal service workers (17.6%) and sales workers (15.37%) with a 3.9 average skill score. The other two groups closely fit the conventional 'white-collar' and 'blue-collar' grouping. Managers and Professionals account for more than 85% of the 'white-collar' workers, sharing a 1.3 average skill score. 98% of 'blue-collar' workers are male, mostly working as technicians and trades workers, managers, and machinery operators and drivers, and as laborers.

Fig 5 suggests a noticeable difference in the distribution of Skill Score across the three clusters. Most 'white-collar' workers have higher levels of knowledge or formal training. 'Pink-collar' workers have lower levels. The skill level for blue-collar workers appears to be more evenly distributed, reflecting the increased skills and training required of many workers in these industries today. Locations of employment also differed by cluster. More 'white-collar' workers worked in the central business district of Brisbane; 'pink-' and 'blue-collar' workers' worked more in suburban areas.

## Travel behaviour analysis of three types of workers

Figs 6 and 7 shows commuting behaviours of the three clusters based on respondents' journey-to-work trips only. Private motor vehicle dominated commuting across the three groups: 81.4% of all commutes made by 'white-collar' workers; 86% for 'pink-collar' workers, and 92.6% for 'blue-collar' workers. In terms of median trip distances by private motor vehicle, the white-collar workers tended to commute shorter distances (13.7 km) than blue-collar workers (18.4 km) whilst pink-collar workers have the shortest average driving distance (12.3 km). The proportion of white-collar workers commuting by public transportation was almost three times greater than the blue-collar workers (14.8% versus 5.7%). However, blue-collar workers tended to travel longer distances when commuting by public transport when compared to

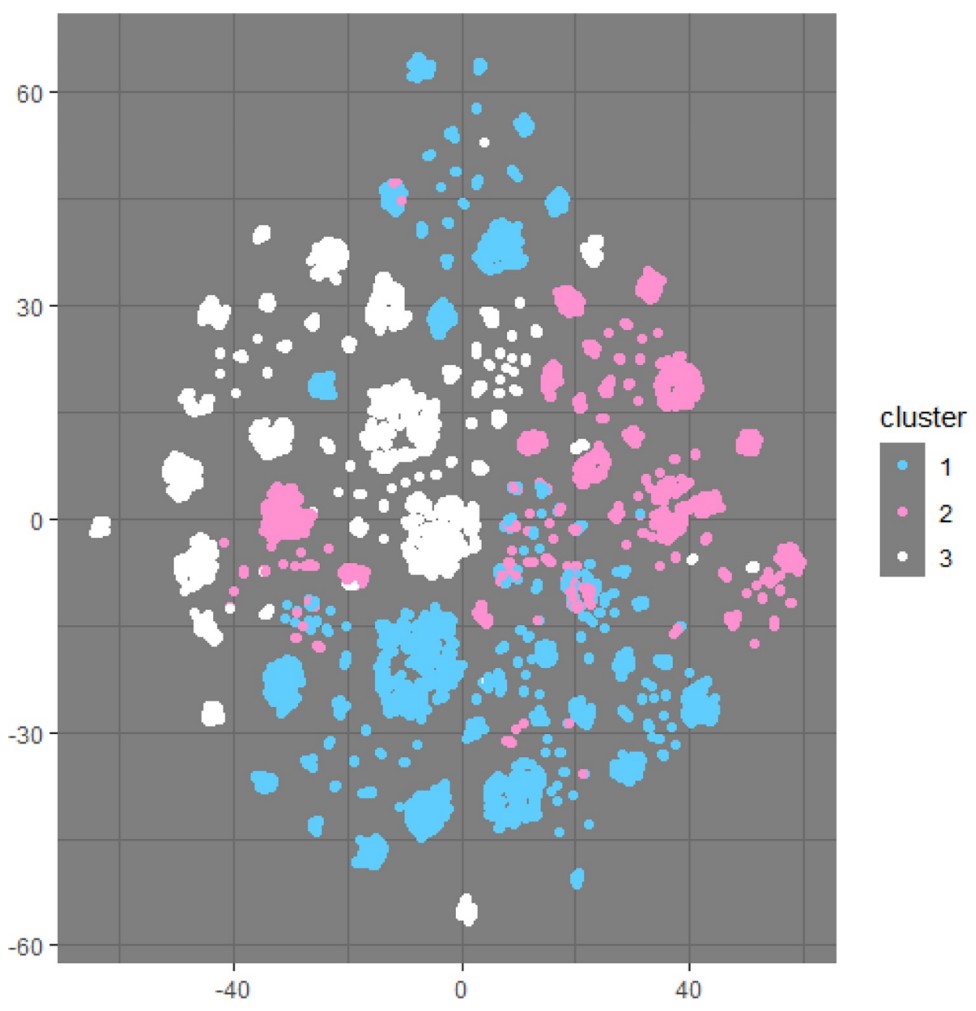

**Fig 3. The t-SNE representation for three clusters.**

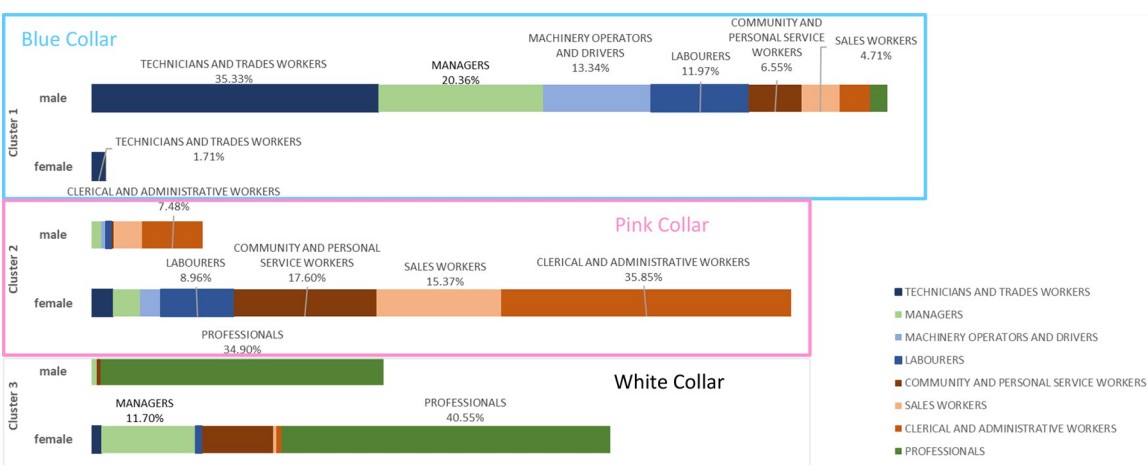

**Fig 4. Component analysis of three clusters by gender and occupation.**

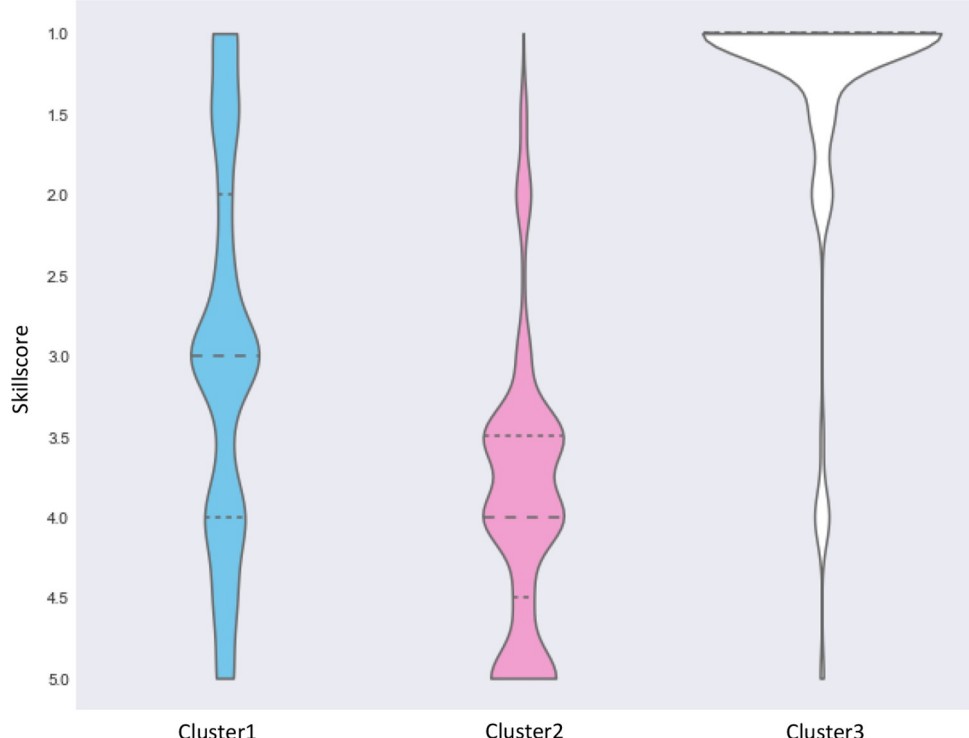

**Fig 5. Distribution of Skill Score across the three clusters.**

white-collar workers (25.8 km vs 19.2 km, respectively). Pink-collar workers have a modest share of commutes made by public transport (11.5%) but have a similar median trip distance for these trips to 'white collar' workers (21.1 km vs. 19.2km, respectively). White-collar workers were mostly likely to use active transport modes, and their trip distances were almost twice

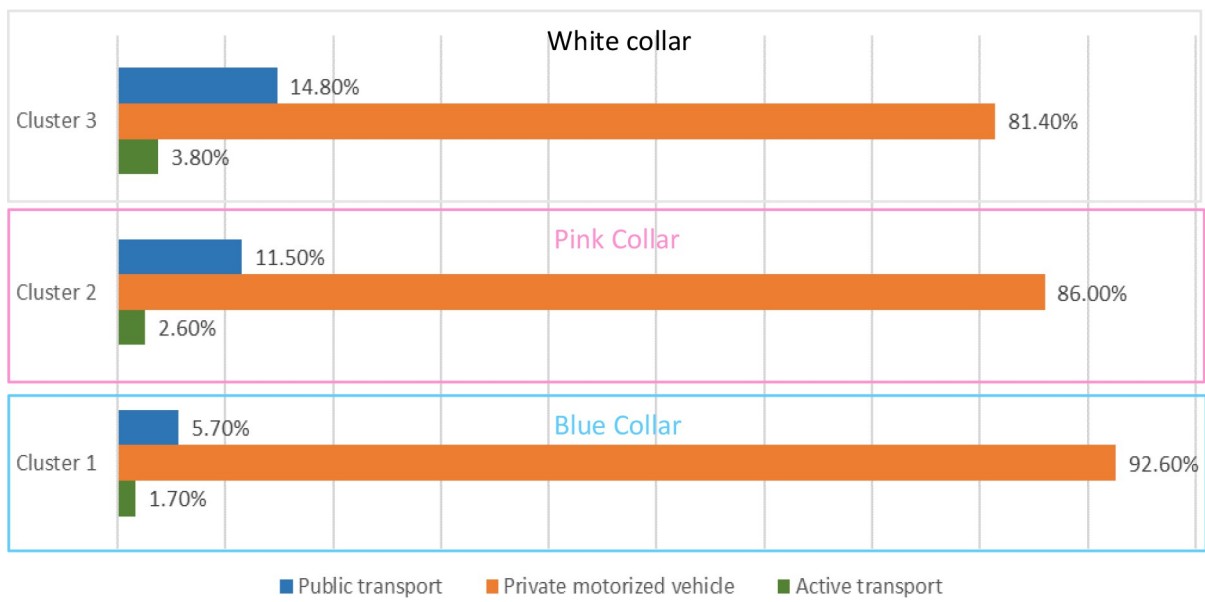

**Fig 6. Mode share comparison across the three clusters.**

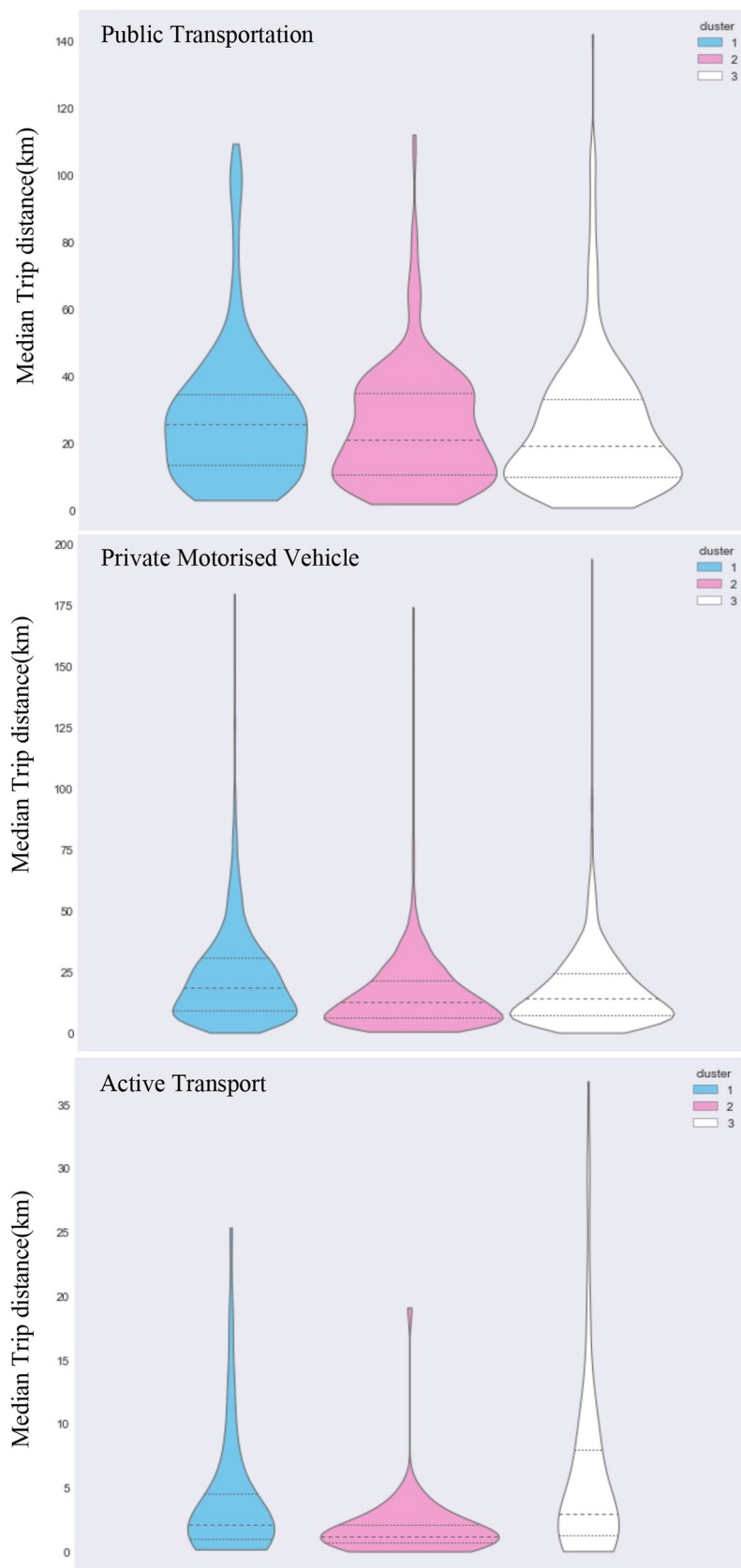

**Fig 7. Median trip distance distribution by mode for the three clusters.**

as far as the 'pink-collar' group; this would have health benefit implications for the different groups [50,51]. These results are consistent with previous research on Australian households, showing that women often take work closer to home in the suburbs to be closer to schooling and childcare [52]. The results suggest that commuting behaviours between the three clusters are statistically dissimilar.

## Discussion

There is a need to continue to improve the accuracy of city-region transport models, to generate more robust demand forecasting. This paper has made a number of small contributions towards this task; the analysis being just one modest step in identifying potential improvements. Firstly, the results confirm that those who deductively identified and included pink-collar market segments in urban transport model elsewhere were on the right track. Second, the paper has shown that inductive clustering techniques can be used to explore commute types at a number of levels and see what is happening as different groupings are aggregated. This approach is not just theoretically more robust, the analysis shows it can directly improve market segmentation choices. The approach also allows each city to go and explore its own datasets and come up with reasonable and justifiable 'solutions' for the market segmentation problem, regardless of its peculiar employment market. What may work for New York may be very different to that of Brisbane, and different again in Honolulu. As household travel survey data are increasingly made available freely via open data portals, the methods developed in this paper should help planners, modellers and researchers across a litany of cities improve the accuracy of their models.

There are limitations and one must be cautious in interpreting the results. Female-dominated or male-dominated clusters, and their travel behaviour, should not be conflated with the commuting behaviour of men or women, per se. There is nuance needed in taking these results through and operationalising them in mode choice models. The results suggest that the conventional blue/white collar segmentation strategy for transport models may fail to sufficiently represent female workers' travel behaviour appropriately, at least in Australia. However, this conjecture should be tested by comparing the predictive ability, by gender, of models using a blue/white collar segmentation to that of models using the blue/white/pink collar segments identified in this paper.

There are also other data-driven approaches that could take this work further. Latent class mode choice studies could provide a richer picture and additional improvements. There also appears significant worth in further exploring the travel characteristics of the two distinct female-dominated clusters of commuters' travel identified in Table 3 when there were eight clusters (k = 8). This suggests that pink-collar workers are potentially two key groups (we label these, '*aqua-collar*' and '*purple-collar*' commuters?). When computing power and transport modelling advances to the point where a higher level of disaggregation is possible without pushing out model run times, it may be of value to include such differentiation in the market segmentation problem. At the present time, however, the benefits of doing so in model accuracy are likely out-weighed by the problems of model complexity and run-time. Finally, methodological and cultural change will continue in the future. Transport researchers may look back at this paper and admonish it for omitting a litany of new variables, and for its own conscious and unconscious decisions. That is to be both expected, and encouraged.

## Author Contributions

**Conceptualization:** Yiping Yan, Matthew Burke.

**Data curation:** Yiping Yan, Abraham Leung.

**Formal analysis:** Yiping Yan.

**Funding acquisition:** Matthew Burke.

**Investigation:** Yiping Yan.

**Methodology:** Yiping Yan, Abraham Leung, James McBroom.

**Resources:** Matthew Burke.

**Supervision:** Abraham Leung, Matthew Burke, James McBroom.

**Validation:** Yiping Yan.

**Visualization:** Yiping Yan, Abraham Leung.

**Writing – original draft:** Yiping Yan.

**Writing – review & editing:** Yiping Yan, Abraham Leung, Matthew Burke.

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
