## [Decision Letter · Decision Letter 0]

10 Jul 2023

PONE-D-23-09999Understanding and including ‘pink-collar’ workers to overcome the sublimation of women in travel demand models and forecastingPLOS ONE

Dear Dr. Yan,

Thank you for submitting your manuscript to PLOS ONE. After careful consideration, we feel that it has merit but does not fully meet PLOS ONE’s publication criteria as it currently stands. Therefore, we invite you to submit a revised version of the manuscript that addresses the points raised during the review process.

Perform a comprehensive revision for typography errors and punctuation.

Review the manuscript's structure to align the title with its content and aims.

State the advantages and limitations of the methodology

Provide access to the database

Clarify and explain all technical terms and define their meaning when necessary.

Add a conclusion

The bibliography must be updated.

We look forward to receiving your revised manuscript.

Kind regards,

Humberto Merritt, PhD

Academic Editor

PLOS ONE

Journal Requirements:

2. Please ensure that you have specified (1) whether consent was informed and (2) what type you obtained (for instance, written or verbal, and if verbal, how it was documented and witnessed). If your study included minors, state whether you obtained consent from parents or guardians. If the need for consent was waived by the ethics committee, please include this information.

4. We noted in your submission details that a portion of your manuscript may have been presented or published elsewhere. [Yes, this article is a research article, which has been previously submitted and presented at the Australasian Transport Research Forum.] Please clarify whether this conference proceeding or publication was peer-reviewed and formally published. If this work was previously peer-reviewed and published, in the cover letter please provide the reason that this work does not constitute dual publication and should be included in the current manuscript.

Additional Editor Comments:

Although the paper's aim is challenging, it must be reordered and completed. The analysis has been performed correctly, but some parts need a deeper explanation. The cluster analysis is based only on a few variables, but other relevant variables are not referred to in the study, such as race, age, etc.

On the other hand, the paper's core argumentation is unclear, and the claim of the present segmentation is "inductive," whereas the older ones were "deductive," which is challenged by several inconsistencies throughout the manuscript. In addition, the study suggests that conventional blue/white collar segmentation strategy for transport models may fail to represent female workers' travel behavior in Australia adequately and that a deterministic split between women and men should be an essential component of the segmentation process, but these claims must be substantiated with more evidence.

The empirical analysis lacks a more comprehensive justification. The usage of the clustering approach needs a more detailed explanation to mention the method's advantages and drawbacks. The database must be provided, or a link where it can be accessed.

There must be a careful usage of technical and methodological terms to avoid misleading conceptualizations. In particular, a thoughtful revision of the conceptual meanings, such as sublimation, CBD, etc., must be carried out. Similarly, all subjective opinions and qualifications must be supported with documented evidence, preferably using a summarizing list or table.

Many references are more than ten years old.

The manuscript lacks a section dealing with the main conclusions of the study. As a result, there are no comments on the study's policy implications and a proposal for future research.

Finally, since the manuscript has already been presented at a congress, a statement must be added to mention the paper's character as the updated version of that submission.

Due to these conditions, the editor's decision for the submitted manuscript is that it needs a major revision of the whole paper.

To this effect, you have 45 days to address these comments, requiring resubmission of the corrected version to Journal PLOS ONE.

I encourage you to attend to the reviewers' comments fully. In particular, I ask you to pay attention to the following issues.

Required changes

Perform a comprehensive revision for typography errors and punctuation.

Review the manuscript's structure to align the title with its content and aims.

State the advantages and limitations of the methodology

Provide access to the database

Clarify and explain all technical terms and define their meaning when necessary.

Add a conclusion

The bibliography must be updated.

Reviewers' comments:

Reviewer's Responses to Questions

**Comments to the Author**

1. Is the manuscript technically sound, and do the data support the conclusions?

Reviewer #1: Partly

Reviewer #2: Partly

2. Has the statistical analysis been performed appropriately and rigorously? 

Reviewer #1: Yes

Reviewer #2: Yes

3. Have the authors made all data underlying the findings in their manuscript fully available?

Reviewer #1: Yes

Reviewer #2: No

4. Is the manuscript presented in an intelligible fashion and written in standard English?

Reviewer #1: Yes

Reviewer #2: Yes

5. Review Comments to the Author

Reviewer #1: The paper addresses a very specific topic which is the segmentation of respondents of the Brisbane standard transport surveys. The authors could present their findings in a broader international context.

The methods are correctly applied, but the research process needs to be reordered and completed.

The statistical analysis been performed correctly, but some steps merit a deeper explanation.

Only the results of the analysis are presented. Data sources are mentioned and they are probably available, pero they do not mention it.

The paper refers several times along the paper to the initial motivation related to updating the old division of trips between blue and white-collar workers. Once at the beginning is enough. Another general comment is about the type of trips considered in the analysis; the segmentation analysis referred only to commuting trips, but it is not clearly said, that only those trips are selected. The issue is not only to survey workers but for their trip to work.

The introduction makes a very detailed reference to ANZSCO, which is explained later in section 2.2. The introduction could only refer to the problem and then describe it.

Chapter 3 includes the methodology. However, it needs to start to describe and define the characteristics of the SEQTS and BSTM. Only people very familiar with the current practice in that specific city/area could value the findings and the analysis. By explaining the survey and modelling context, the analysis could be better understood and valued. International readers need to highlight the importance of the outputs to other cities/countries. Clarify which means active transport. If walking and cycling or something else, like multimodal trips with some stages not motorized.

The results should help not only consultants and transport planners in Brisbane but in other contexts. To that end, the authors should reflect on which findings are general or only valid for local conditions.

The clusters are based on a number of variables, including the trip length for each mode. It would be nice to comment on the text which is the average/median distance for all. I understand that trip distance refers to the Origin-Destination route. It surprises me that PT trips have longer itineraries than private cars. Normally people use cars because they live/work in more distant places. If not, then the household economic level should be included as a variable. Which means CBD? Clarify.

Reviewer #2: General comments:

This study can make a useful contribution: I can see how three clusters, one of which largely identifies pink collar workers, are an improvement over two clusters, largely representing blue and white collar workers, respectively. However, much of the paper’s core argumentation is unclear and/or illogical.

In particular, the claim that older segmentations were “deductive”, based on modelers’ (presumptively biased) hunches about appropriate segmentation variables, while the present one is “inductive”, “let[ting the] data speak for itself” (line 62), seems quite overdrawn to me. Biases are not excluded from the putatively data-driven approach of the present study either. First of all (and through no fault of the authors), previous biases and hunches – about what is appropriate and important to measure – have determined what variables are even *available* to the cluster analysis. Second, the *present* analysts’ hunches about what variables are relevant to identifying segments that “place different importance on different aspects of service” (line 241) are governing *this* analysis as much as *prior* analysts’ hunches governed *theirs*. For example, the present cluster analysis includes a number of variables, but apparently does not include race, sexual orientation, income, presence of children, ages of children, number of elders, number of wage-earners, nor transit pass ownership, as well as spatial and attitudinal characteristics (as mentioned by the authors). Some those variables may not be available in the dataset (see the “first of all” point above), but some presumably are available, and were consciously or unconsciously rejected by the authors – on their hunch – as being insufficiently relevant.

And by the way, if you want to be even *more* data-driven, why not use latent class mode choice and other travel behavior models, to identify the *segments that best differentiate the segment-specific coefficients* of the outcome model? As it is, the authors are still performing a deterministic segmentation in an entirely separate step from the travel behavior model to which they see the clusters eventually being applied, with no guarantee that this is the *best* segmentation for the purposes of identifying segments that “place different importance on different aspects of service”.

I recognize that a latent class choice model is a different paper than this one, and I’m not insisting that this one be replaced with a latent class choice model. I am simply saying that the authors need to be more objective and logically consistent about their characterizations of the alternative approaches that have been taken. Best to suggest that analysts in each era probably do the best job they can of objectively considering what’s important, but that no analysts, including the present ones, are bias-free, and it is quite possible (even likely) that future analysts will criticize the present study for omitting important variables that society later recognizes as critical – in the same way that the present study is criticizing previous studies for doing the same.

In short, please remove discussion of what I see as a false dichotomy between deductive and inductive approaches, and do not claim (line 181) that “Inductive approaches are better suited for this type of analysis as they remove most potential researcher bias.”

lines 408-410, “In applied terms, the results provide strong empirical evidence that the conventional blue/white collar segmentation strategy for transport models fails to sufficiently represent female workers’ travel behaviour appropriately, at least in Australia”: Not really. If included at all, the statement needs to be softened to something like, “The results suggest that the conventional blue/white collar segmentation strategy for transport models may fail to sufficiently represent female workers’ travel behaviour appropriately, at least in Australia. However, this conjecture should be tested by comparing the predictive ability, by sex, of models using a blue/ white collar segmentation to that of models using the blue/white/pink collar segments identified in this paper.”

But I’m also uneasy about the multiple references to it being *women’s* travel behavior that is being overlooked or misrepresented (e.g. lines 39, 54-55, 164-166, 410). The white-collar segment of this study is 64% women/36% men, so in this important segment, women and men are still being smushed together – as are people with different numbers of children, seniors, adults, workers, and drivers in the household, etc. If it’s truly *women’s* travel behavior that needs to be clearly distinguished, then a deterministic split between women and men should be an essential component of the segmentation process, regardless of what other variables might also be considered. Instead though, I gather that it may be *pink-collar workers’* travel behavior (most of whom are women, but far from all women are pink-collar) that is being subsumed under cruder segmentation systems. I agree that we don’t want to do that, but let’s call the problem what it is, and be precise about what the study is actually doing. The paper issues a caveat along these lines at lines 403 – 406, but more or less ignores that caveat for the preceding 95% of the paper. How *should* someone “tak[e] these results through and operationalis[e] them in mode choice models”? It seems like a pretty important question for the paper to answer, in view of its admonitory tone throughout.

Also, the paper needs a thorough edit for grammar, punctuation, and typos. Time doesn’t permit an exhaustive list, but one example is at line 142, “the forebear of that work”. “Forebear” means “ancestor”, whereas the context requires it to say “descendent”.

Specific comments:

In the title, “sublimation” (and, at line 165, “sublimated”) is not really the right word, unless there is some Australian distinction that is unknown to this native American English speaker (and therefore to a large share of this paper’s potential audience). I recommend “subsumation”, consistent with the paper’s use of “subsumed” at line 54.

line 20, “Conventional transport models tend to segment commuters as either blue or white collar workers”: I would say, “often” (or, really, “sometimes”), since that particular segmentation is not at all common in my experience. It’s not even clear if it’s any longer common in Australia – the paper (line 53-54) refers to “key Australian transport models of the past”, but what about the present? Please confirm that references 6-10 provide “lots” of examples of the *current* use of that segmentation. OK, lines 141-142 give one current example – are there others? Enough to justify use of the phrase “tend to”? Even if so, the contention should still be qualified with “*Australian* transport models”.

line 57: Although not all new things are novel, all novel things are, by definition, new, so it suffices to say “novel typology”, rather than “new and novel typology”.

line 214: I was a little surprised that the SEQTS data represented an average per-person trip rate of 2.8 trips per day. In the US it is 3.4 trips per person per day (https://nhts.ornl.gov/assets/2017_nhts_summary_travel_trends.pdf), and I wouldn’t have thought that the Southeast Queensland region would be so different.

lines 270-275: This was rather confusingly written. I would like to rewrite it to say, “Most simply, Gower distance is computed as the average of partial dissimilarities across individuals, in which each numeric-valued feature is standardized, and the distance is calculated as the average of all feature-specific distances. However, if variables are of mixed (qualitative as well as numeric) types, partial dissimilarity is calculated differently. For numeric features, it is computed as the ratio between absolute difference of observations Xi and Xj and maximum range observed from all individuals (thus scaling all dissimilarities to lie between 0 for identical, and 1 for maximally dissimilar):” If this characterization is correct, it would be much clearer to me for it to be presented this way.

lines 305-307, “each cluster is dominated by certain industries and occupations sharing similar commute mode preferences and similar work trip distances”: For a given cluster, it is not clear how similar mode preferences and commute distances are within industry and occupation, since each cluster comprises an undifferentiated bundle of differing shares of industries and occupations. I suggest just focusing on distinct differences across clusters.

Table 4: The white collar and pink collar column heads appear to be reversed, based on Figure 3 and statements in the text. I.e. the information in the second column pertains to the pink collar cluster and the third column describes the white collar cluster. And regarding the variables from CBD down, are we talking about commute trips specifically? Please clarify; also in Table 3.

Figure 3: Many readers (including this one) will not have the foggiest idea what the “t-Distributed Stochastic Neighbour Embedding(t-SNE) algorithm” is, and accordingly have no way of interpreting Figure 3. Please at least explain what the two axes represent, and preferably also give a brief explanation of what the t-SNE algorithm is and does.

6. PLOS authors have the option to publish the peer review history of their article (what does this mean?). If published, this will include your full peer review and any attached files.

Reviewer #1: No

Reviewer #2: No

---

## [Author Response · Author response to Decision Letter 0]

14 Feb 2024

Thank you for the useful comments and suggestions. All the comments that were suggested by the reviewer have been addressed the "Table of Revisions" file. Please refer to that file for changes made according to each comment.

---

## [Editor Report · Decision Letter 1]

11 Mar 2024

Understanding and including ‘pink-collar’ workers in employment-based travel demand models

PONE-D-23-09999R1

Dear Dr. Yiping Yan,

We’re pleased to inform you that your manuscript has been judged scientifically suitable for publication and will be formally accepted for publication once it meets all outstanding technical requirements.

Kind regards,

Humberto Merritt, PhD

Academic Editor

PLOS ONE

Additional Editor Comments (optional):

After carefully revising the changes, corrections and adequations made to the previously submitted version of the manuscript titled “Understanding and including ‘pink-collar’ workers in employment-based travel demand models,” I confirm that this version has met the academic requirements and comments posed by the reviewers, and now fulfills the academic and quality criteria for publication in PLOS ONE.

So, please follow the submission instructions provided by PLOS ONE for further editorial directions.
---

## [Editor Report · Acceptance letter]

29 Mar 2024

PONE-D-23-09999R1 

PLOS ONE

Dear Dr. Yan, 

I'm pleased to inform you that your manuscript has been deemed suitable for publication in PLOS ONE. Congratulations! Your manuscript is now being handed over to our production team.

Kind regards, 

on behalf of

Dr. Humberto Merritt 

Academic Editor

PLOS ONE